

# Ocean wave tracing v.1: A numerical solver of the wave ray equations for ocean waves on variable currents at arbitrary depths.

Trygve Halsne[1,2], Kai Haakon Christensen[1,3], Gaute Hope[1], and Øyvind Breivik[1,2]

[1]Norwegian Meteorological Institute, Oslo, Norway
[2]University of Bergen, Bergen, Norway
[3]University of Oslo, Oslo, Norway

**Correspondence:** Trygve Halsne (trygve.halsne@met.no)

**Abstract.** Lateral changes in the group velocity of waves propagating in oceanic or coastal waters causes a deflection in their propagation path. The change in direction is called refraction and can be computed if having information about the ambient current field and/or the bathymetry. We present an open source module for solving the wave ray equations by means of numerical integration in Python v3, which is relevant for ocean applications. The solver is implemented for waves at arbitrary

depths and for an ambient current field with spatio-temporal variability much lower than characteristic wave properties. The wave ray tracing module is implemented in a class structure, and the output is verified against analytical solutions as well as tested for numerical convergence. The solver is accompanied by a set of ancillary functions with the aim of supporting relevant workflows for the user community including data retrieval, transformation, and dissemination, and a number of use examples are provided.

**1   Introduction**

Ambient currents and varying water depth affect the propagation path of ocean waves through refraction and can induce substantial horizontal wave height variability and complex sea states through crossing rays, leading to caustics (Fig. 1) (Holthuijsen, 2007). The linear theory of wave kinematics has been known for almost a century and stems from applying the Wentzel–Kramers–Brillouin approximation (WKB, and sometimes WKBJ, where the last initial refers to Jeffreys) on char-

acteristic wave and current conditions (Kenyon, 1971). That is, the changes in wave amplitude $a$, angular intrinsic frequency $\sigma$, and ambient medium are small on distances on the order of a wave length $\lambda$. Such a treatment is known as the *geometrical optics approximation*, and is applicable in various scientific branches dealing with propagation of wave rays on different frequency scales. The resulting set of equations, typically referred to as the "wave ray equations", have a limited number of analytical solutions such that numerical integration is necessary to designate the propagation path (Kenyon, 1971; Mathiesen,

1987; Johnson, 1947). Such solvers have been available in the ocean wave community since the advent of spectral wave models, but often as part of a large and complex model framework, and not generally available as stand-alone applications.

Recent developments in the ocean modelling community, including assimilation of observations, have led to more realistic ocean model output fields, which in turn have led to an increased interest in wave-current interaction studies (Babanin et al., 2017). Current-induced refraction has often been singled out as the principal mechanism leading to horizontal wave height

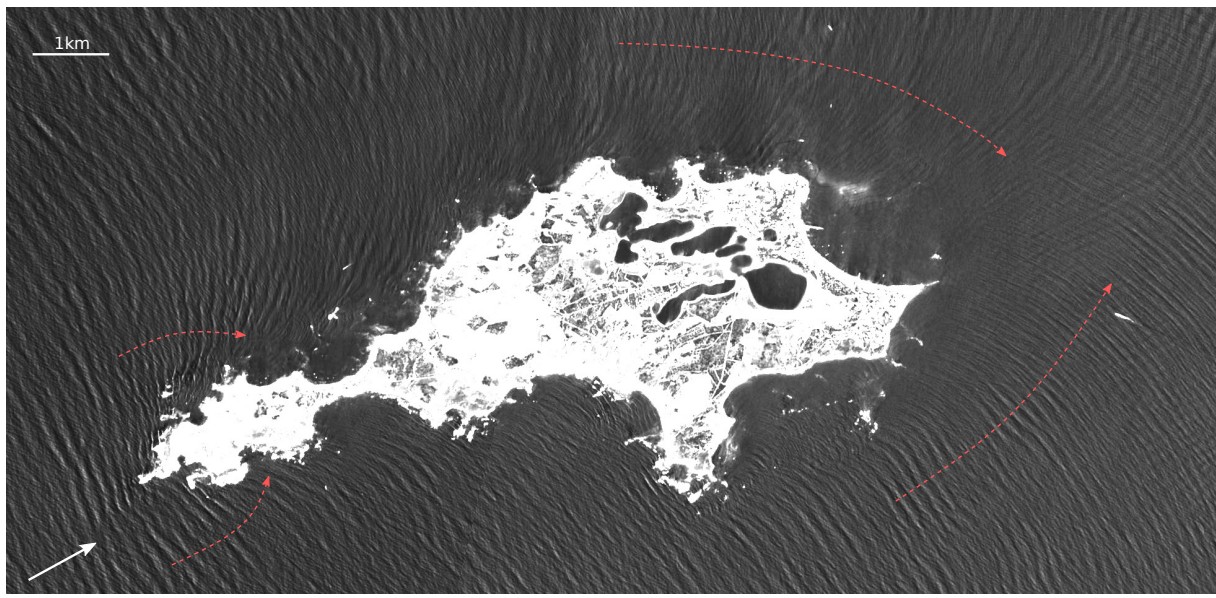

**Figure 1.** Depth refraction of swell against the *Rottnest Island* off the coast of Western Australia depicted by the Copernicus Sentinel-2 mission, processed by ESA, in December 2021. The swell propagates north-eastwards (white arrow), and interacts with the bathymetry when coming close to the island. Red arrows indicate the change in wave propagation direction, which is normal to the wave crest. An area subject to crossing waves is obtained on the east side of the island due to the change in wave propagation direction.

variability at scales between one to several hundred kilometers (e.g. Irvine and Tilley, 1988; Ardhuin et al., 2017, 2012). Thus, a number of recent studies employ wave ray equation solvers in order to quantify the impact of refraction (e.g. Romero et al., 2017, 2020; Ardhuin et al., 2012; Masson, 1996; Bôas et al., 2020; Halsne et al., 2022; Saetra et al., 2021; Sun et al., 2022; Gallet and Young, 2014; Rapizo et al., 2014; Kudryavtsev et al., 2017; Bôas and Young, 2020; Jones, 2000; Segtnan, 2014; Mapp et al., 1985; Wang et al., 1994; Liu et al., 1994). However, such implementations are rarely open to the community. To the

best of our knowledge there exists no open-source solver available in a high level computer language to support such analyses. Furthermore, some of them only focus on deep water where the wave ray equations are simplified since the topographic steering is negligible (e.g. Bôas and Young, 2020; Bôas et al., 2020; Mathiesen, 1987; Kenyon, 1971; Rapizo et al., 2014; Kudryavtsev et al., 2017). However, the joint effect of current- and depth-induced refraction at intermediate depth can be important (Halsne et al., 2022).

The scope of this paper is to present an open-source numerical solver of the wave ray equations implemented in Python. The paper is structured as follows: In Section 2 we present the theoretical background for the geometrical optics approximation of the wave ray equations on ambient currents and in variable depth. The numerical discretization and implementation of the equations and model is given in Section 3. Furthermore, some ancillary functions that support efficient workflows are presented. In Section 4, we compare the numerical solver against analytical solutions and inspect the numerical convergence. A selection





of examples using the ray tracing module, including idealized current fields and output from ocean circulation models, are presented in Section 5. Finally, a brief discussion and some concluding remarks are provided in Section 6.

## 2 Derivation of the wave ray equations

For simplicity, we first derive the wave ray equations in the $x$-direction, and then extend the results to both horizontal directions. We assume linear wave theory such that $ak \ll 1$, where $a$ denotes the wave amplitude and $k = |\mathbf{k}| = |(k_\mathrm{x}, k_\mathrm{y})|$ is the wave

number. When considering the kinematics of wave trains through the geometric optics approximation, it should be emphasized that diffraction is neglected. For a more complete description of the kinematics and dynamics of ocean waves, we refer the reader to Phillips (1977) and Komen et al. (1994).

### 2.1 The one dimensional problem

A plane wave propagating in a slowly varying medium is given by

$$\eta(x,t) = ae^{i\chi}, \tag{1}$$

where $\chi = kx - \sigma t + \delta$ is the phase function. Here $x, t, \delta$ denote position, time and the phase, respectively, and $\sigma$ is the wave angular intrinsic frequency given by the dispersion relation

$$\sigma = \sigma(k,x) = \sqrt{gk \tanh(kd)}, \tag{2}$$

where $d = d(x)$ is the water depth, which we assume to be constant in time. In the presence of an ambient current $U = U(x,t)$,

the absolute wave angular frequency is

$$\omega = \Omega(k,x,t) = \sigma + kU, \tag{3}$$

which is often referred to as the "Doppler shift equation". Considering a phase function $\chi' = kx - \omega t + \delta$ in a frame of reference not moving with the current, we obtain the "conservation of wave crests"

$$\frac{\partial k}{\partial t} + \frac{\partial \omega}{\partial x} = 0. \tag{4}$$

Taking the partial derivative of (3) while keeping $t$ constant, we obtain

$$\frac{\partial \omega}{\partial x} = \frac{\partial \Omega}{\partial k}\frac{\partial k}{\partial x} + \frac{\partial \Omega}{\partial x}, \tag{5}$$





where $\partial\Omega/\partial k = c_{\mathrm{g}} + U$ is the advection velocity, which contains the wave group velocity $c_{\mathrm{g}}$. We define the material (or total) derivative as

$$\frac{d}{dt} = \frac{\partial}{\partial t} + (c_{\mathrm{g}} + U)\frac{\partial}{\partial x}. \tag{6}$$

Thus, advection of a wave group with position $x$ is simply

$$\frac{dx}{dt} = \frac{\partial\Omega}{\partial k} = c_{\mathrm{g}} + U, \tag{7}$$

which is the first of the wave ray equations. The evolution of the wave number $k$ follows by inserting (5) into (4) such that

$$\frac{dk}{dt} = -\frac{\partial\Omega}{\partial x} = -\left(\frac{\partial\sigma}{\partial x} + k\frac{\partial U}{\partial x}\right), \tag{8}$$

which is the second of the wave ray equations. Using the same approach for $\omega$, we get

$$\frac{d\omega}{dt} = \frac{\partial\Omega}{\partial t} = 0. \tag{9}$$

Here, we keep in mind that we consider ambient currents that vary slowly compared with a characteristic wave period. Thus, the absolute wave frequency is constant. Summarized, we obtain the wave ray equations in one horizontal dimension

$$\frac{dx}{dt} = c_{\mathrm{g}} + U, \tag{10}$$

$$\frac{dk}{dt} = -\left(\frac{\partial\sigma}{\partial x} + k\frac{\partial U}{\partial x}\right), \tag{11}$$

$$\frac{d\omega}{dt} = 0. \tag{12}$$

The wave ray equations constitute a set of coupled ordinary differential equations (ODEs) that define a characteristic curve in space and time. They can be solved as an initial value problem if defined with a starting point $x^{n=0} \equiv x(t=0)$ and an initial wave period $T = T^{n=0}$ by using the dispersion relation (2). In deep water, where the wave length $\lambda \ll d/2$, the first term on the right hand side of (11) vanishes since $\tanh(kd) \to 1$ in (2). Under such conditions the evolution of $k$ is only a function of

the horizontal gradients in the ambient current.

## 2.2 The two dimensional problem

In 2D, we denote the position vector $\mathbf{x} = (x, y)$ and the ambient current vector $\mathbf{U} = (U, V)$. We define the horizontal gradient operator

$$\nabla_{\mathrm{h}} \equiv \hat{\mathbf{i}}\frac{\partial}{\partial x} + \hat{\mathbf{j}}\frac{\partial}{\partial y}, \tag{13}$$





where $\hat{\mathbf{i}}, \hat{\mathbf{j}}$ denote the unit vectors for $x, y$, respectively. Now, the absolute angular frequency

$$\omega = \Omega(t, \mathbf{k}, \mathbf{x}) = \sigma + \mathbf{k} \cdot \mathbf{U}(t, \mathbf{x}), \tag{14}$$

and the wave ray equations become

$$\frac{d\mathbf{x}}{dt} = c_{\mathrm{g}} + \mathbf{U}(t, \mathbf{x}) \tag{15}$$

$$\frac{d\mathbf{k}}{dt} = -\nabla_{\mathrm{h}}\sigma - \mathbf{k} \cdot \nabla_h \mathbf{U}(t, \mathbf{x}), \tag{16}$$

$$\frac{d\omega}{dt} = 0. \tag{17}$$

In the context of spectral wave modelling, the dynamical evolution of the wave field is prescribed by the wave action balance equation

$$\frac{\partial N}{\partial t} + \nabla_{\mathrm{h}} \cdot \left(\dot{\mathbf{x}}N\right) + \nabla_k \cdot \left(\dot{\mathbf{k}}N\right) = \frac{S}{\sigma}. \tag{18}$$

Here, $N \equiv E/\sigma$, is the wave action density, which is a conserved quantity in the presence of currents (Bretherton and Garrett,

1968). The wave action density contain the wave variance density $E$, which is $\propto a^2$. The right hand side of (18) represents sources and sinks of wave action. The wave number gradient operator

$$\nabla_{\mathrm{k}} \equiv \hat{\mathbf{i}}\frac{\partial}{\partial k_{\mathrm{x}}} + \hat{\mathbf{j}}\frac{\partial}{\partial k_{\mathrm{y}}}. \tag{19}$$

The wave ray equations are integrated in (18) by the terms denoted with overdots, where

$$\dot{\mathbf{x}} = \frac{d\mathbf{x}}{dt}, \tag{20}$$

$$\dot{\mathbf{k}} = \frac{d\mathbf{k}}{dt}. \tag{21}$$

There is thus a connection between the wave field dynamics and kinematics where $\dot{\mathbf{x}}$ represents the advection of wave action in physical space, and $\dot{\mathbf{k}}$ the refraction ("advection" in $\mathbf{k}$-space). The wave action balance equation (18) is solved by third generation spectral wave models, but then discretized either by wave number $k$ or frequency $f$, and direction $\theta$ (Komen et al., 1994).

## 3 Numerical implementation

### 3.1 Finite-difference discretization

The wave ray equations (15)–(16) are well suited for numerical integration. The `ocean_wave_tracing` module offers two finite-difference numerical schemes: a 4th order Runge-Kutta and a Forward Euler scheme through its `solver` method. For readability, the latter is here used to present the discretization of the wave ray equations. The advection (15) becomes

$$x_{(l,j)}^{n+1} = x_{(l,j)}^n + \Delta t\, f_{adv}\left(x_{(l,j)}^n, y_{(l,j)}^n, k_{(l,j)}^n, k_{x,(l,j)}^n, U_{(l,j)}^n\right), \tag{22}$$

$$y_{(l,j)}^{n+1} = y_{(l,j)}^n + \Delta t\, f_{adv}\left(x_{(l,j)}^n, y_{(l,j)}^n, k_{(l,j)}^n, k_{y,(l,j)}^n, V_{(l,j)}^n\right). \tag{23}$$





Here, $n$ denotes the discrete time index, with $n = 0, 1, \ldots, N$ and $\Delta t = t_{n+1} - t_n$. Discrete horizontal indices are given by $l = 0, 1, \ldots, N_x$, $j = 0, 1, \ldots, N_y$, and $\Delta x = x_{l+1} - x_l$, $\Delta y = y_{l+1} - y_l$. The $f_{\mathrm{adv}}$ is a function of the group velocity and ambient current, and becomes (skipping time and horizontal indices for readability)


$$f_{\mathrm{adv}}(x, y, k, k_{\mathrm{x}}, U) = \begin{cases} c_{\mathrm{g}}(k, d[x,y]) \frac{k_{\mathrm{x}}}{k} + U(x,y), & \text{in } x\text{-direction,} \\ c_{\mathrm{g}}(k, d[x,y]) \frac{k_{\mathrm{y}}}{k} + U(x,y), & \text{in } y\text{-direction.} \end{cases} \tag{24}$$

The evolution in wave number (16) becomes

$$k_{x,(l,j)}^{n+1} = k_{x,(l,j)}^{n} + \Delta t\, f_{\mathrm{conc}}\left(k_{x,(l,j)}^{n}, k_{y,(l,j)}^{n}, \frac{\partial}{\partial x}\sigma(k_{l,j}^{n}, d_{l,j}), \frac{\partial}{\partial x}U_{l,j}^{n}, \frac{\partial}{\partial x}V_{l,j}^{n}\right), \tag{25}$$

$$k_{y,(l,j)}^{n+1} = k_{y,(l,j)}^{n} + \Delta t\, f_{\mathrm{conc}}\left(k_{x,(l,j)}^{n}, k_{y,(l,j)}^{n}, \frac{\partial}{\partial y}\sigma(k_{l,j}^{n}, d_{l,j}), \frac{\partial}{\partial y}U_{l,j}^{n}, \frac{\partial}{\partial y}V_{l,j}^{n}\right). \tag{26}$$

Here, $f_{\mathrm{conc}}$ is a function of the horizontal derivatives $\sigma$ and the $\mathbf{U}$. The abbreviation conc refers to the *concertina effect* as it is
called by Ardhuin et al. (2017). For the horizontal derivatives, we use a central difference scheme, such that $f_{\mathrm{conc}}$ becomes

$$f_{\mathrm{conc}}(x, y, k_{\mathrm{x}}, k_{\mathrm{y}}, U, V) = \begin{cases} -\frac{\sigma_{l+1,j}^{n} - \sigma_{l-1,j}^{n}}{2\Delta x} - k_{x,(l,j)}^{n} \frac{U_{l+1,j}^{n} - U_{l-1,j}^{n}}{2\Delta x} - k_{y,(l,j)}^{n} \frac{V_{l+1,j}^{n} - V_{l-1,j}^{n}}{2\Delta x}, & \text{in } x\text{-direction,} \\ -\frac{\sigma_{l,j+1}^{n} - \sigma_{l,j-1}^{n}}{2\Delta y} - k_{x,(l,j)}^{n} \frac{U_{l,j+1}^{n} - U_{l,j-1}^{n}}{2\Delta y} - k_{y,(l,j)}^{n} \frac{V_{l,j+1}^{n} - V_{l,j-1}^{n}}{2\Delta y}, & \text{in } y\text{-direction.} \end{cases} \tag{27}$$

## 3.2 Stability condition

A constraint for hyperbolic equations in finite-difference numerical schemes is the Courant–Friedrichs–Lewy (CFL) condition which for a process with advection velocity $W$ demands that the non-dimensional Courant number


$$C \equiv W \frac{\Delta t}{\Delta r} \leq 1, \tag{28}$$

where $\Delta r = \sqrt{\Delta x^2 + \Delta y^2}$. If $C > 1$, the process will advect a distance larger than the grid point resolution over a period $\Delta t$, leading to instabilities in the numerical solution. A dedicated method, `check_CFL`, is implemented in the `Wave_tracing` class and added in the `set_initial_condition` method (Fig. 2). The Courant number is written to the log file as

$$C_{\mathrm{logfile}} = \begin{cases} \text{info,} & \text{if } C \leq 1, \\ \text{warning,} & \text{if } C > 1. \end{cases} \tag{29}$$

The advection velocity (the absolute group velocity as seen from a fixed point) in (28) is implemented as

$$W = \max(|\mathbf{U}|) + \max(\mathbf{c}_{\mathrm{g}}^{n=0}), \tag{30}$$

which is a good proxy for the magnitude of the maximum advection speed. It may, however, exceed $W$ for $n > 0$ for waves starting in shallow water and propagating towards deeper water. In the `check_CFL`, $\Delta r = \min(\Delta x, \Delta y)$.





### 3.3 Model simulation workflow

The wave ray equations are implemented in Python3 in the `ocean_wave_tracing` module available on GitHub at https://github.com/hevgyrt/ocean_wave_tracing under a GPL v.3 license. It is based on common native Python libraries and open source projects. Key open source projects include *numpy* (numerical Python – https://numpy.org/), *matplotlib* (https://matplotlib.org/), and *xarray* (https://docs.xarray.dev/en/stable/). The latter library is a large project, which has become a de-facto standard in geophysical sciences for analyzing and dealing with multi-dimensional data. The wave ray tracing tool is a class instance

and the `Wave_tracing` object contains multiple auxiliary methods before and after performing the numerical integration. Here, we will focus on the workflow, input fields, implementation, and the ancillary methods enclosing the wave ray tracing `solver` method.

#### 3.3.1 Operating conditions

A set of fixed conditions are specified for the `ocean_wave_tracing` module. The most important include:

– The model domain must be rectangular and in Cartesian coordinates with a uniform horizontal resolution in each direction.

– Units follow the SI system with length scale units of meters m and seconds s. The angular units are radians rad. Wave propagation direction $\theta$ follow a right handed coordinate system with $\theta = 0$ being parallel to the $x$-axis and propagating in the positive $x$-direction.

– Variable names, structure, and its metadata are to a large extent based on the Climate and Forecast (CF) metadata convention (https://cfconventions.org/).

#### 3.3.2 Ray tracing model initialization

A flowchart of the model simulation workflow is given in Fig. 2 and an associated code example in Alg. 1. Firstly, a wave ray tracing object `Wave_tracing` is initialized by an `__init__` method. The input variables define the ambient conditions,

and include

– the ambient current `U`, `V` = **U**,

– the bathymetry `depth` (optional),

– the boundaries `X0, XN, Y0, YN` and horizontal resolution `dx, dy` of the domain,

– the number of time steps `nt` and total duration time for wave propagation `T`,

– the number of wave rays `nb_wave_rays`.



---

**Algorithm 1** Generic workflow code example

---

```python
import numpy as np
import maplotlib.pyplot as plt
from ocean_wave_tracing import Wave_tracing

# Defining some properties of the medium
nx = 100; ny = 100 # number of grid points in x- and y-direction
x = np.linspace(0,2000,nx) # size x-domain [m]
y = np.linspace(0,3500,ny) # size y-domain [m]
T = 250 # simulation time [s]
U=np.zeros((nx,ny))
U[nx//2:,:]=1

# Define a wave tracing object
wt = Wave_tracing(U=U,V=np.zeros((ny,nx)),
                  nx=nx, ny=ny, nt=150,T=T,
                  dx=x[1]-x[0],dy=y[1]-y[0],
                  nb_wave_rays=20,
                  domain_X0=x[0], domain_XN=x[-1],
                  domain_Y0=y[0], domain_YN=y[-1],
                  )

# Set initial conditions
wt.set_initial_condition(wave_period=10,
                         theta0=np.pi/8)
# Solve
wt.solve()
```

---





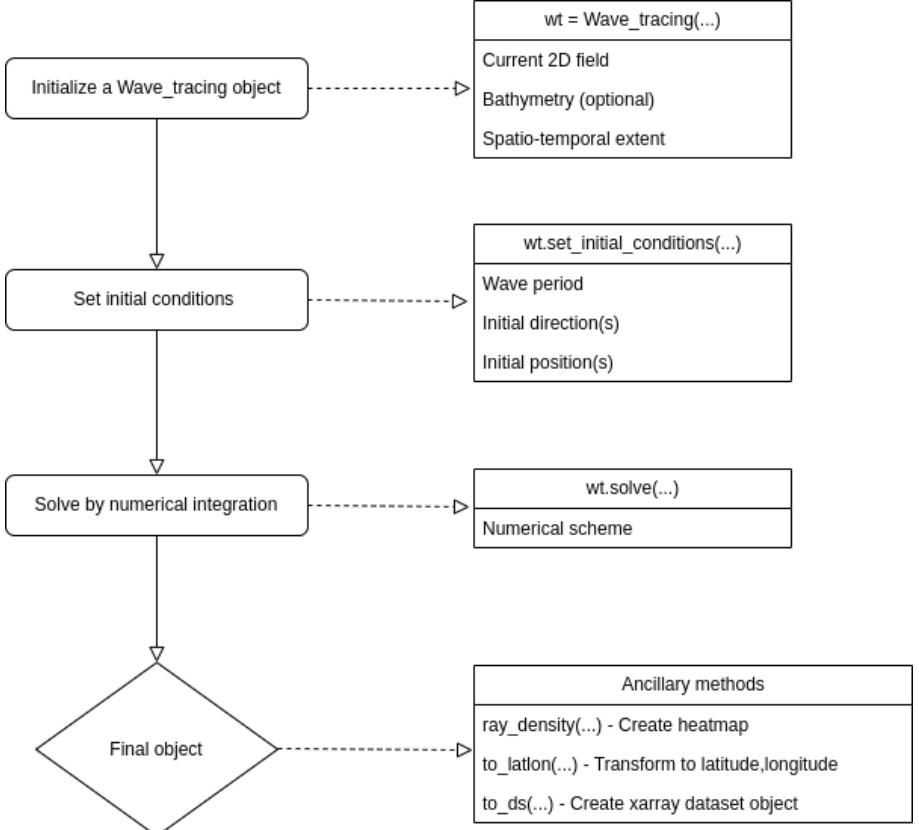

**Figure 2.** Flowchart of the workflow from initializing a ray tracing object to solving the wave ray tracing equations. Left column denote the most important steps in the workflow, and the right column highlight the most important parameters and supporting methods under each step.

The current is allowed to vary in time by setting `temporal_evolution=True`, but it is up to the user to make sure that $\mathbf{U}(t, \mathbf{x})$ is not violating (17) by $\partial \mathbf{U}/\partial t \simeq 0$. If the bathymetry is not specified, the model assumes deep water waves and sets a fixed uniform depth at $10^5$ m. Depth values are defined as positive, implying that negative values will be treated as land if both negative and positive values are present through a dedicated bathymetry checker (`check_bathymetry`), which is invoked
within `__init__`. Furthermore, the input velocity field is also checked, and *xarray* datasets are created for the bathymetry and velocity field as class variables following the CF convention.

### 3.3.3 Setting the initial conditions

Before the numerical integration, initial conditions for the ODEs are specified in a dedicated `set_initial_condition()` method (Alg. 1, Fig. 2). Here the initial wave period $T^{n=0}$, wave propagation direction $\theta = \theta^{n=0}$, and initial position $\mathbf{r}(t = 170 \quad 0, \mathbf{x}) = (x^{n=0}, y^{n=0})$ are specified. By utilizing the rectangular model domain, the initial position can most easily be given as one of the sides of the domain, i.e. `top, bottom, left` or `right`, where `left` is default (see Alg. 1). In such cases, the

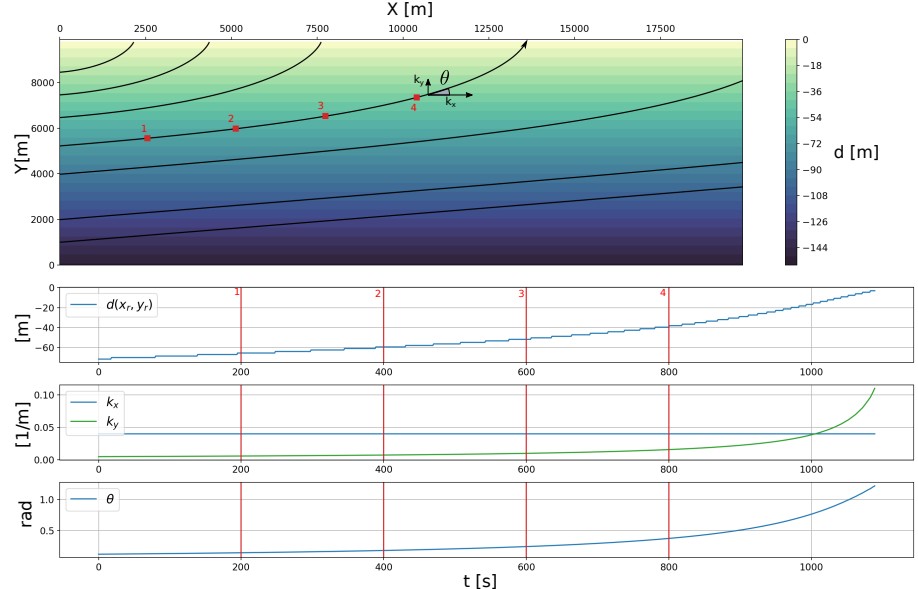

**Figure 3.** A conceptual figure highlighting the workflow strategy of the model. Here, depth refraction of a $T = 10$ s period wave propagating from left with an initial angle $\theta^{n=0} = 0.1$ rad is shown for seven different rays (black lines). The propagation path for each wave ray is computed simultaneously through vectorization. The lower panels denote the change in depth, evolution in $\mathbf{k}$ and corresponding wave propagation direction, $\theta$, in time along one of the rays. As expected, the wave rays deflect towards shallower regions due to the increase in the $k_\mathrm{y}$ wave number component.

number of wave rays is spread uniformly on the selected boundary. Another option is to specify initial grid points `ipx, ipy` for each wave ray. Similarly, $\theta = \theta^{n=0}$ can also be specified for each ray, or a single uniform direction can be given for all rays. Such examples are provided later.

The model is solved for a single wave frequency, dictated by the initial wave period $T^{n=0}$. The wave number $k$ is retrieved from $T^{n=0}$ by (2), which in intermediate depths requires an iterative solver. Using the approximation by Eckart (1952), the error in $k$ is less than 5% (Holthuijsen, 2007).

### 3.3.4 Numerical integration

Numerical integration of (22)–(27) is initiated by invoking the `solver` method. Here, $\nabla_h \mathbf{U}$ is computed prior to the integra-
tion using the *numpy gradient* method. The integration is performed iteratively in a Lagrangian sense by computing the next position $\mathbf{r}^{n+1}$ from the current position $\mathbf{r}^n$ for each wave ray. Thus, the `solver` keeps track of the horizontal indices $l, j$ for every time step and for each wave ray in the model domain. Hence, the numerical integration for the wave rays can follow a vectorized approach, which is conceptually visualized in Fig. 3. For a given position $\mathbf{r}$, the properties of the ambient medium, i.e., the current and bathymetry, is selected using a nearest neighbor approach.





Even though $\nabla_h \mathbf{U}$ is static for each model field, $\nabla_h \sigma$ in (27) must be computed for each iteration $n$ since the wave number $k$ evolves in time. Furthermore, for each iteration $n$, the wave propagation direction $\theta^n$ is computed from $\mathbf{k}^n$ using the *numpy atan2* function.

After a successful call of the `solve()` function, the `Wave_tracing` object have populated its class variables for the wave rays being (`ray_x`, `ray_y`), (`ray_kx`, `ray_ky`), `ray_k`, `ray_theta`, `ray_cg`, (`ray_U`, `ray_V`), which are the

horizontal position vector, wave number vector, wave number, wave propagation direction, wave group velocity, and ambient current vector, respectively. All of the aforementioned class variables have dimensions `number_of_wave_rays`$\times N$.

The numerical scheme used in the `solver` method is configurable by the user, and the default is a 4th-order Runge Kutta scheme. That is, the numerical scheme is generic and detached from the wave ray equations. The schemes are available in a separate utility function `util_solvers`, which contains (currently two) numerical schemes which are defined Python

classes in a hierarchy with a generic ODE solver as the top node. That is, each sub-class has its own *advance* method, which corresponds to the numerical scheme. This approach is to a large extent built on material from Langtangen (2016). Furthermore, the `util_solvers` also contain the advection and wave number evolution functions (24) and (27), respectively.

### 3.4 Ancillary methods and testing

#### 3.4.1 Ancillary functions

Ancillary functions include methods which are considered useful for the user community. In the current version it includes four methods, i.e. three within the `Wave_tracing` object and one outside the object.

The method outside the `Wave_tracing` object is targeted for data preparation before model initialization. It is not strictly a Python method, but a generic workflow for data retrieval. More specifically, since the ray tracing model is focused on ocean currents and bathymetry, it is natural to exploit variable fields from ocean circulation models. It is common for oceano-

graphic centers to disseminate model results under a free and open data policy, and to enable the Open source Project for a Network Data Access Protocol (OPeNDAP–https://www.opendap.org/) on the data distribution server (e.g. THREDDS–https://www.unidata.ucar.edu/software/tds/current/ or HYRAX–https://www.opendap.org/software/hyrax-data-server). The OPeN-DAP enables spatio-temporal subsetting to be carried out serverside and thus avoids the problem of downloading huge amounts of data prior to use. Such user-defined subsets can be accessed directly by data streaming by using common netCDF4 readers

(https://www.unidata.ucar.edu/software/netcdf/), which among others is available in *xarray*. The ancillary method, or work-flow, is provided in the jupyter-notebook `extract_ocean_model_data.ipynb`. Here, the user can plot and check the user-defined area and temporal extent prior to writing the subset to disk, or initiating the `Wave_tracing` object directly. It is common for ocean circulation models to have output variable fields with hourly temporal resolution such that $\mathbf{U}(t, \mathbf{x})$ is unlikely to violate (17). However, it is up to the user to understand the limitations of the model if simulating wave rays for very

long shallow water waves like tsunamis and tidal waves.

The first of the three class methods within the `Wave_tracing` object is a transformation method from projection coordinates to latitude and longitude values, which is called `to_latlon()` (see Fig. 2). That is, when using ocean circulation model





field variables as input data, it is not readily possible to compare the `Wave_tracing` output with other sources of data since ocean model field variables are most often in a specific projection. In this context, using latitude and longitude coordinates

are often much more convenient. The method requires the *proj4*-string of the ocean model domain and performs coordinate transformation using the *pyproj* (https://pyproj4.github.io/pyproj/stable/) library in Python. Even if not required, it is common that the *proj4*-string is listed in the `grid_mapping` variable in a CF-compliant ocean model dataset.

The second ancillary function is based on the wave ray density method by Rapizo et al. (2014) and is called `ray_density()`. It computes the relative number of wave rays within user-defined grid boxes, which can be considered proportional to the wave

height and thus the horizontal wave height variability. The method returns a 2D grid and the associated ray density variable.

The third method takes care of converting all the characteristic `Wave_tracing` class variables into a *xarray* dataset, including latitude and longitude if the *proj4* string is given as input. The method is called `to_ds()`. The output *xarray* dataset follow the CF convention for metadata. Thus, the data can utilize all the functionality within *xarray*, including the plotting and writing data to disk. Examples of using all the methods listed above will be shown later in Section 5.

### 3.4.2   Tests

The `ocean_wave_tracing` repository is equipped with unit-tests written in the framework of Pythons `pytest`. Unit tests are tailored for the methods within and used by the `Wave_tracing` class, and typically checks the numerical implementation against known solutions. For instance, the computation of wave celerity for deep and shallow water are tested against analytical solutions.

For integration tests, a set of example scripts running the entire chain of operations are embedded in the `test` folder. Such tests are also implicitly inherent in the scripts provided in the `notebooks` and `verification` folders, since these notebooks run the entire chain. Moreover, continuous integration tests are embedded in the repository utilizing the *poetry* https://python-poetry.org/ project.

## 4   Model validation

Here we verify the output of the `Wave_tracing` solver against analytical solutions for idealized cases for depth- and current-induced refraction. Model differences are given as the absolute relative difference between the analytical solution A and the numerical model solution B for an arbitrary variable $z$ as

$$\Delta(z) = \left| \frac{z_A - z_B}{z_A} \right| \times 100, \tag{31}$$

given in the units of percentage.

### 4.1   Snell's law

When only considering the bathymetry, Snell's law

$$\frac{\sin(\phi_1)}{\sin(\phi_2)} = \frac{c_1}{c_2}, \tag{32}$$





applies for parallel depth contours (see Note 7A Holthuijsen, 2007, p.207). Here, subscripts $1, 2$ indicate the properties of the wave and medium before and after being transmitted through an interface, which here are lines of different bathymetry, and $c$

is the phase speed. The $\phi_1$ denote the incidence angle between the wave ray and the normal to the interface, and $\phi_2$ is the angle of refraction after the interaction.

In the presence of ambient currents, Snell's law can be written for a horizontally sheared current $V = V(x)$ (Kenyon, 1971)

$$\sin(\phi_2) = \frac{\sin(\phi_1)}{(1 - \frac{V}{c_1}\sin(\phi_1))^2}. \tag{33}$$

Verification of the wave ray tracing model results against (32) and (33) are shown in the upper and lower panels of Fig. 4,

respectively. For the idealized bathymetry, the wave ray tracing was performed for a shallow water wave with wave length $\lambda = 10000$ m propagating towards a stepwise shallower region. Here, $\Delta\phi_2$ was computed for each new depth regime (upper panel). For the horizontally sheared current(lower panel Fig. 4), a $T = 10$ s period wave on deep water propagated through the current field

$$V(X) = \begin{cases} 0 & \text{if } X < 2000 \text{ m}, \\ 2, \text{m s}^{-1} & \text{if } X \geq 2000 \text{ m}. \end{cases} \tag{34}$$

The relative differences in both the idealized bathymetry and horizontally sheared current cases listed above were $\Delta(\theta_2) \sim 10^{-1}\%$ (Fig. 4). The script producing Fig. 4 and computing the analytical results is available as a jupyter-notebook under `verification/snells_law.ipynb`.

## 4.2 Wave deflection

For deep water waves, there is a direct relation between wave ray curvature and the vertical vorticity (henceforth vorticity)

$\zeta = \partial v/\partial x - \partial u/\partial y$ (Dysthe, 2001):

$$\nu = \frac{\zeta}{c_{\text{g}}}, \tag{35}$$

valid for $|\mathbf{U}|/c_{\text{g}} \ll 1$. Here, positive vorticity will deflect a wave to the left relative to their wave propagation direction, and to the right for negative vorticity. The ratio with the wave group velocity also entail that shorter waves will deflect more compared with longer waves.

An approximate wave deflection angle can be computed from (35) by adding a characteristic $\zeta = \zeta_0$ and length scale $l$ such that (Gallet and Young, 2014)

$$\theta_\nu \simeq \frac{\zeta_0 l}{c_{\text{g}}}. \tag{36}$$

We use the idealized horizontally sheared current

$$U(X,Y) = \begin{cases} 0, & \text{if } X < 2500 \text{ m}, \\ 3\alpha, & \text{if } X \geq 2500 \text{ m} \end{cases} \tag{37}$$



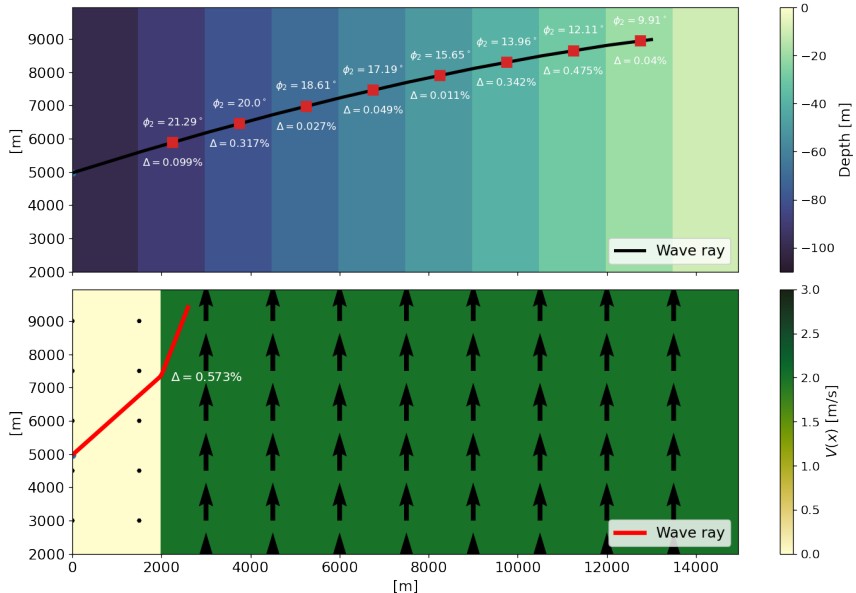

**Figure 4.** Verifying the analytical solutions of Snell's law against the wave ray tracing solver for cases with idealized bathymetry [Eq. (32)] (upper panel) and a shear current [Eq. (33)] (lower panel). The relative difference in $\Delta\theta_2$ [Eq. (31)] are given as insert text for both cases.

where $\alpha$ increase linearly from $\alpha = 0$ at $y = 0$ to $\alpha = 1\,\mathrm{m\,s^{-1}}$ at $y = Y$. An assessment of $\theta_\nu$ for a $T = 10$ s period deep water wave propagated through (37) is shown in Fig. 5. Here, the lower panel solution also uses (37), but with a minus sign in front of $\alpha$. The velocity profile yield a constant $\zeta$. Relative differences between the model and analytical solution are $\Delta(\theta_\nu) \sim 10^{-1}$ %. The difference is a sum of the numerical errors together with the approximate equality in (36). Furthermore, the difference between the simulation of the negative and positive vorticity $\zeta$ is also due to the advection of the current. Furthermore, the deflection direction for negative and positive $\zeta$ is readily seen in Fig. 5. The full analysis is available in the `verification/wave_deflection.ipynb` notebook in the GitHub repository.

### 4.3 Numerical convergence

The numerical convergence for decreasing values of the CFL number $C$ is tested for the conservation of absolute frequency $\omega$ (17). For the idealized case of a deep water wave propagating in the $x$ direction from a region with $U = 0$ to a region with an opposing current $U = -1\,\mathrm{m\,s^{-1}}$, (17) requires

$$\omega = \sigma + kU = \mathrm{const.} = \omega_0, \tag{38}$$

where subscript 0 denotes the region with $U = 0$. For deep water, the phase speed $c = \sigma/k$ such that (38) can be rearranged and

$$k = \frac{k_0 c_0}{U + c}. \tag{39}$$



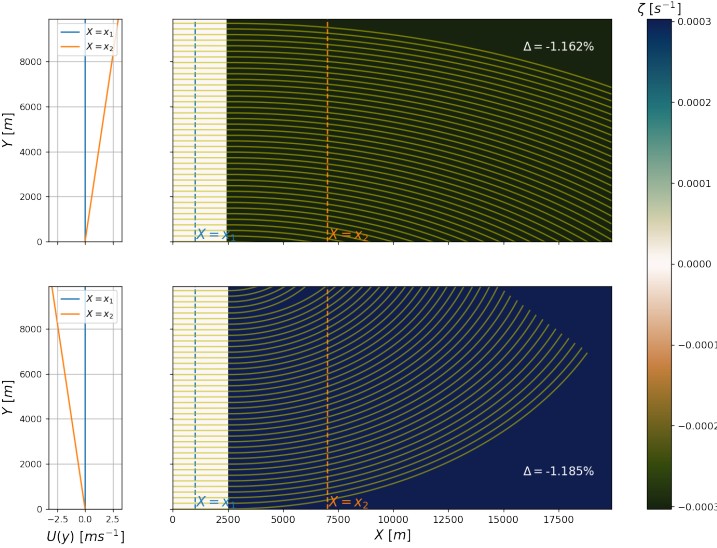

**Figure 5.** Verifying the approximate solutions of wave deflection (36) against the wave ray tracing output for cases with idealized negative (upper panels) and positive vorticity $\zeta$ (lower panels). The associated velocity field $\mathbf{U} = (U(x,y),0)$ at $X = x_1, x_2$ is given in the left column plots. The relative difference $\Delta\theta_\nu$ [Eq. (31)] using $l = 2500$ m are given for both cases (see insert text). Yellow lines denote the wave rays.

In our example, $k = 0.046$ m⁻¹ for a $T = 10$ s period wave propagating into the region $U = -1\,\mathrm{m\,s^{-1}}$. The numerical convergence for (17) is shown in Fig. 6. Here, the error $\Delta\omega$ decreases with decreasing $C$ due to increasing number of time steps $N$. The error does not decrease monotonically, however, since $k$ must be solved sufficiently many times within the region where $\partial U/\partial x \neq 0$ to obtain its correct value. Nevertheless, the solution converges to $k$ (see $k_{\mathrm{an}}$ in Fig. 6) with decreasing $C$. The test on the numerical convergence is available in the GitHub repository in the `verification/numerical_convergence_omega.ipy` notebook.

## 5 Examples of usage

Here we provide some use examples of the wave ray tracing model, which include simulations under idealized current and bathymetry fields and ambient conditions retrieved from an ocean circulation model. The code for running the tool is similar to the generic example given in Alg. 1, however with different ambient and initial conditions. The idealized current fields are part of the repository as a netCDF4 file and reproducible in the `notebooks/create_idealized_current_and_bathymetry.ipy` Here, we also emphasize characteristic workflows including specifying different initial conditions as well as utilizing the ancillary functions described in Section 3.4.1.



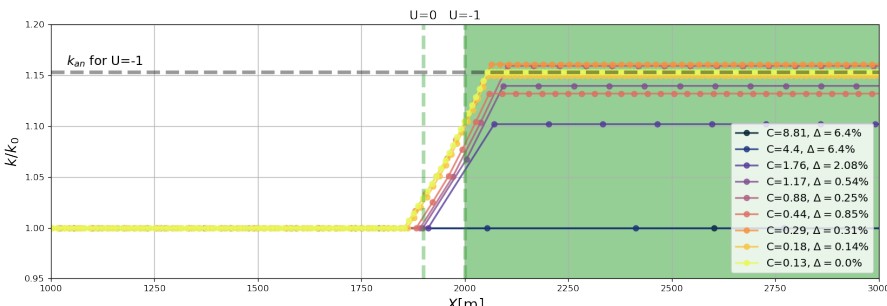

**Figure 6.** The numerical convergence tested for conservation of absolute frequency $\omega$ (17). The convergence is tested for an increasing number of discrete time indices $n$, with an accompanied decreasing Courant number $C$. Here, a domain with velocity field $U = 0$ for $X < 2000$ and $U = -1$ for $X >= 2000$, where $\Delta X = 100$ m (see dashed vertical lines). The analytical value $k_{an}$ is obtained from (39). The relative error $\Delta = (\omega_0 - \omega)/\omega_0$ decreases with $C$. The results here are obtained by using the RK4 scheme, but similar results are obtained using the FE (not shown).

## 5.1 Idealized cases

Cases with depth-induced refraction are shown in Fig. 7. Here the idealized cases show the expected veering of wave rays

towards shallower regions when the deep water limit $\lambda/2 \gg d$ is no longer applicable. The examples also show how the initial position $\mathbf{r}^{n=0}$ can be set differently using the different sides of the domain (i.e. `left` and `bottom` in Figs. 7a,c,d) and from a single point with initial propagation angle uniformly distributed in a sector (Fig. 7b).

     Cases with current-induced refraction in deep water are shown in Fig. 8. Here examples of wave trains both following and opposing a horizontally sheared current are provided in Figs. 8a and b, respectively. The ambient current causes areas of

converging and crossing wave rays, which are known as caustics or focal points. Furthermore, an example of waves propagating through an idealized oceanic eddy is shown in Fig. 8c,d.

     The joint effect of current- and depth-induced refraction at intermediate depth is shown in Fig. 9. Here, the `ray_density` method is used to highlight the different focal points obtained in deep water and when the waves are also influenced by the bathymetry.

All the examples listed above are available in the `notebooks/idealized_examples.ipynb`.

## 5.2 Ocean model output

Examples of surface currents and bathymetry are extracted from the operational coastal ocean circulation model at the Norwegian Meteorological Institute, and wave tracing simulations, are shown for different regions in Fig. 10. Here, the `to_latlon()` method have been used together with the `to_ds()` in order to visualize the output on a georeferenced map. The upper right

panel denote the refraction due to currents and bathymetry (red rays), compared with bathymetry only (yellow rays). There are clear differences between the wave rays with and without currents. The current field used here spanned four model output time

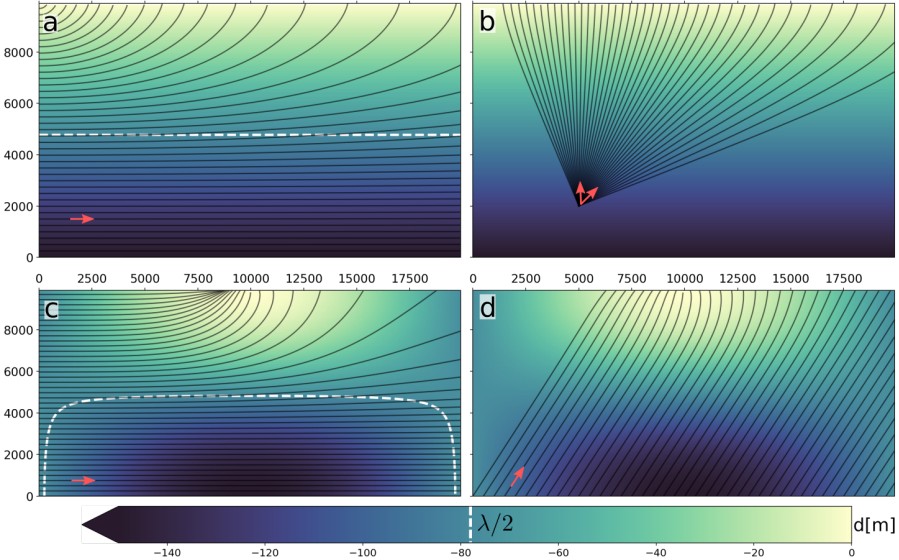

**Figure 7.** Examples of depth refraction of a long crested 10 s period wave using various initial positions and initial propagation directions (red arrows), for two different depth profiles (upper and lower row). Waves with $\frac{\lambda}{2} \gg d$ ($\lambda = 156$ m on deep water, white contour lines), will not "feel" the bottom and thus not be refracted.

steps with hourly temporal resolution. The lower panels of Fig. 10 show how the wave kinematics are affected by a barotropic tidal current under two characteristic cycles. In the lower left panel, the tidal current gives rise to a focal point and crossing wave rays. Cases similar to the two latter was investigated in Halsne et al. (2022), comparing the results with output from a spectral wave model [Eq. (18)]. The examples provided here are available in the `notebooks/ocean_model_example.ipynb`.

The famous textbook-example of trapped waves in the Agulhas current east of South-Africa is shown in Fig. 11. Here, the wave tracing simulations used the surface current from ESAs GlobCurrent project http://globcurrent.ifremer.fr/. The particular point in time for the simulation is the same as used in Kudryavtsev et al. (2017) (i.e. 2016-01-04, see their Figs. 14–15), however here with an apparently coarser horizontal resolution in the current forcing.

## 6 Discussion and concluding remarks

We have presented an Python-based, open source, finite difference ray tracing model for arbitrary depths and with ambient currents. The `Wave_tracing` module has been tested and verified against analytical solutions as well as tested for numerical convergence. The solver comes with a set of ancillary functions aimed at supporting relevant workflows for data retrieval, transformation, visualisation in the scientific community as well as being compatible with the standardized Climate and Forecast (CF) metadata convention. Such workflows have been documented and are available in the repository as examples for the end users.

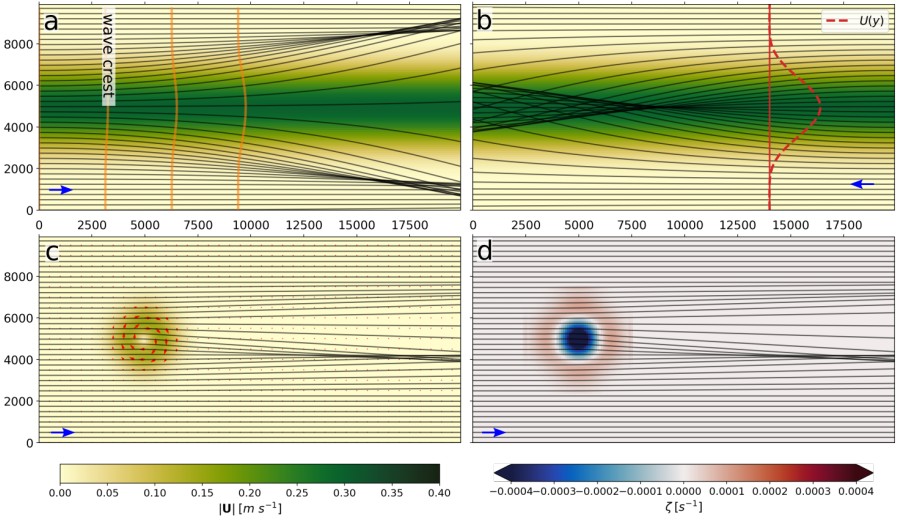

**Figure 8.** Examples of current-induced refraction under different flow regimes and initial propagation directions (blue lines). Panel a) denote the evolution of the wave crest (orange lines) as it rides along a current jet (see current profile $U(y)$ in b). In panel b) the current jet is opposing the waves inducing focal points in the middle of the jet. Panel c) show how a characteristic current whirl impact the wave propagation paths and and d) denote the relation between deflection angle and the vorticity, $\zeta$.

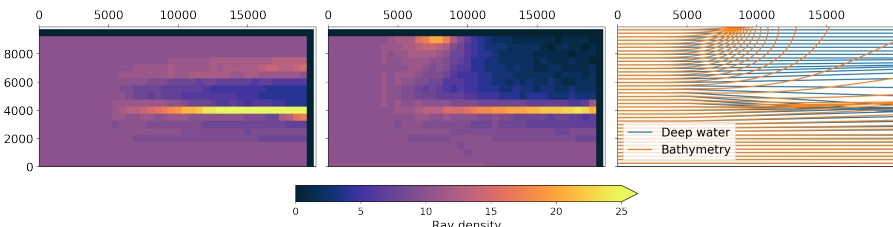

**Figure 9.** The impact of depth and current refraction on horizontal wave height variability as seen through wave ray density plots. Left panel show results using the wave ray density method for a current whirl on deep water (see Fig. 8c). Middle panel show the impact on wave ray density for the same current whirl but on intermediate depths, i.e. adding the bathymetry in Fig. 7c. Right panel denote the difference in the wave ray paths between the two cases.

To the best of our knowledge, no such modeling tool is openly available in a high level computing language despite its usefulness for the investigation and quantification of the impact of ambient currents and bathymetry on the wave field (e.g. Romero et al., 2017, 2020; Ardhuin et al., 2012; Masson, 1996; Bôas et al., 2020; Halsne et al., 2022; Saetra et al., 2021; Sun

et al., 2022; Gallet and Young, 2014; Rapizo et al., 2014; Kudryavtsev et al., 2017; Bôas and Young, 2020).

The solver is applicable to waves in finite depth, which is in contrast to previously reported models which handle only current-induced refraction in deep water (e.g. Bôas and Young, 2020; Bôas et al., 2020; Mathiesen, 1987; Kenyon, 1971;

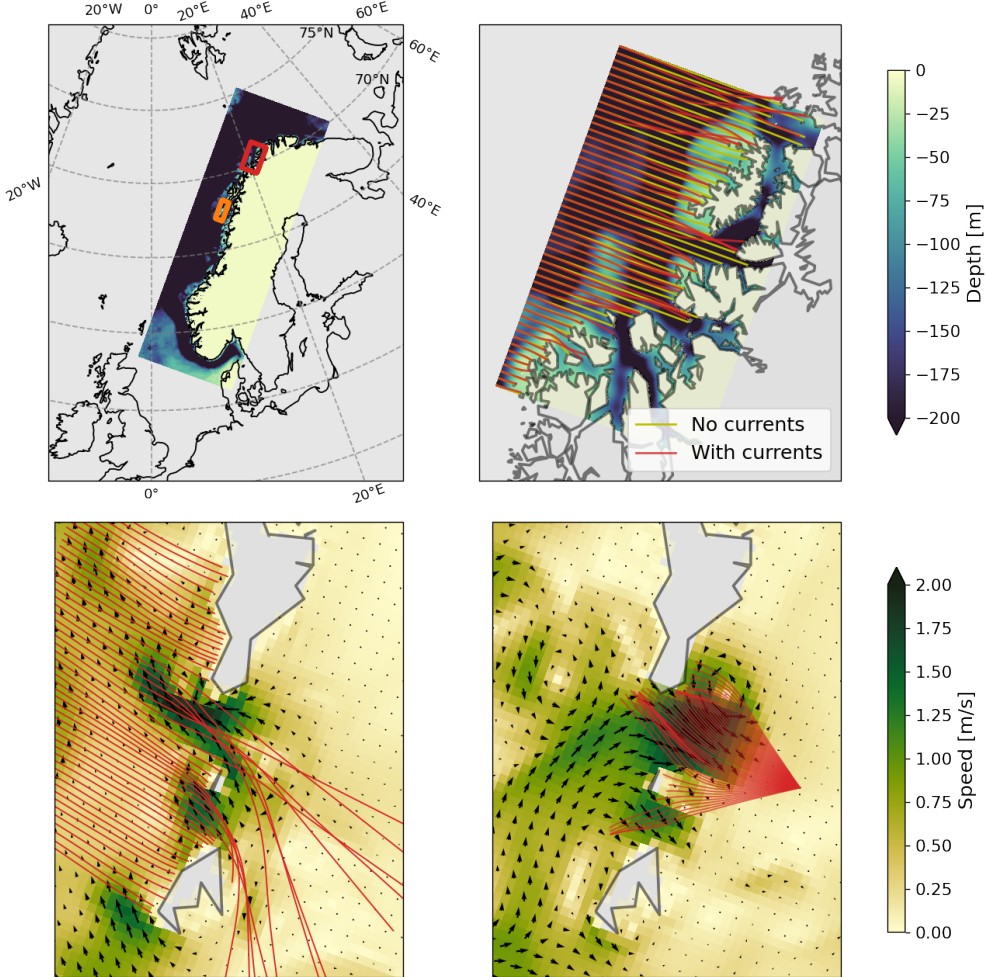

**Figure 10.** The impact of currents and bathymetry on wave propagation paths using current and bathymetry fields from an 800 m resolution ROMS model in Northern Norway (upper left). Upper right plot (subset in the red rectangle) show how time varying surface currents (i.e. four model timesteps) impact the wave propagation paths for a $T = 10$ s period wave (red rays) when compared to refraction due to bathymetry only (yellow rays). Bottom panels (subset in the orange rectangle) show the impact of a tidal current on the wave propagation paths for a $T = 10$ s period wave (lower left) and a $T = 7$ s period wave (lower right). Here, arrows denote the direciton of the ambient current.

Rapizo et al., 2014; Kudryavtsev et al., 2017). For intermediate depths, the joint effect of current- and depth-induced refraction can be very important for the wave height variability (Fig. 9). Such an example was highlighted in Halsne et al. (2022), where
the combined refraction of wave rays due to the ambient current and bathymetry caused focusing which led to a significant increase in the local wave height around a caustic (see their Fig. 11).

The `ocean_wave_tracing` module does not support wave reflection. Such processes are complex and can be added later on. This means that wave rays can essentially propagate through land and out of the model domain. Such effects are, however,



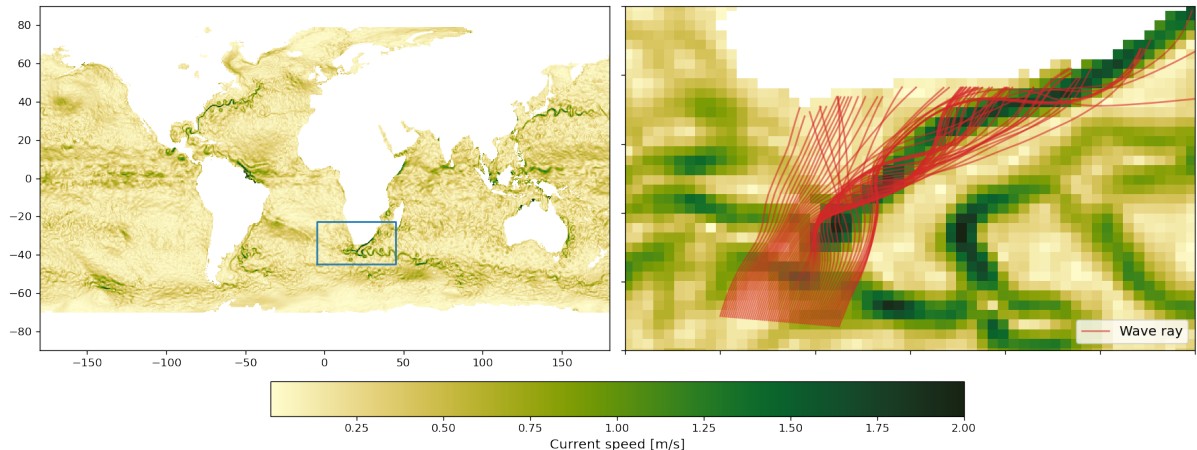

**Figure 11.** Focusing of wave rays by the Agulhas current. Here, surface current data from the GlobCurrent project is used from 2016-01-04 (left panel). The data is originally given in spherical latitude/longitude coordinates, but here approximated in the area of interest to equidistant Cartesian coordinates with 22 km and 28 km spatial resolution in $x$- and $y$-direction, respectively. The famous trapping of wave rays within the branch of the Agulhas current is shown in the right panel using a $T = 10$ s period wave. The duration of the run was 62 hours, and consequently 18 consecutive model output times (temporal resolution of 3 hours) were used.

circumvented by using *numpys'* masked arrays and not-a-number NaN values. For instance in the `bathymetry_checker`, negative bathymetry values will be treated as land and set to a *numpy* NaNs. When plotting, NaN values does not appear when plotting. Furthermore, masked values are often standard in ocean circulation models, and thus wave rays "stop" when entering land grid points (see Fig. 10).

Solving the ray equation using a high level language such as Python gives added execution time and memory usage compared to lower level languages. However, execution times are normally in the order of $10^1$ seconds, but increases for long simulations, which requires an number of time steps `nt`. It is most certainly possible to further speed up the code by utilizing other modules and by making the code base more dense in terms of reducing the amount of code. However, the objective of the wave ray tracing tool described here is neither to create a substitute to wave models nor to optimize it for large and/or long simulations. It is rather to provide a framework that is easy to understand and simple to run. In addition, a comprehensible code base makes the tool suitable for further development by other contributors. Furthermore, best-practices like vectorization have been used in order to speed up the solver, without loss of general readability of the code.

*Code availability.* The source code is available on https://github.com/hevgyrt/ocean_wave_tracing a under a GPL v.3 license with DOI https://zenodo.org/badge/latestdoi/362749576 .





*Author contributions.* TH designed and implemented the software including the formal analysis and model validation, and wrote the manuscript draft. GH supported the development and GitHub integration. GH, KHC, and ØB reviewed and edited the manuscript.

*Competing interests.* The authors declare that they have no conflict of interest.

*Acknowledgements.* This research was partly funded by the Research Council of Norway through the project MATNOC (grant no. 308796). TH and ØB are grateful for additional support from the Research Council of Norway through the StormRisk project (grant no. 300608). TH would like to dedicate this work to Dr. Hans Petter Langtangen post mortem.





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
