# Peer review of "Ocean wave tracing v.1: A numerical solver of the wave ray equations for ocean waves on variable currents at arbitrary depths."

_Geoscientific Model Development, 2023_

## Author Comment (AC1)

We would like to thank both reviewers and the editors for taking the time to review our manuscript and for providing us with valuable feedback. We appreciate your help in improving our work, and we have revised the article following your suggestions.

In addition to the reviewer comments, we also took the opportunity to correct some typos as well as adjusting some of the equations in order to be more precise.

Detailed responses to each of the reviewers comments are found below. Comments from reviewers are in *italic*, and our responses are in **boldface**. Instead of listing new and removed text in the manuscript, a "latexdiff" document will be added along with the revision, which denotes the differences between the old and revised manuscript.

On behalf of the authors,
Trygve Halsne

**Reviewer #1: Dr. Leonel Romero**

*The authors present a numerical solver of the ray equations for surface gravity waves accounting for slowly varying depth and currents. I believe the code should be of value to the community. Below I provide a list with minor suggestions.*

*Specific comments*

*Line 5 reads "… spatio-temporal variability much lower than characteristic wave properties". I suggest being more specific.*

**Thanks for the comment. To be more specific we removed the part referred to by the reviewer, and added that we follow the WKB approximation. The changes are shown in the abstract in the latexdiff document.**

*8: What data retrieval and transformation are you referring to?*

**Thanks for pointing this out. We have changed the text to specifically refer to data retrieval utilizing OPeNDAP and that transformation refers to coordinate transformation. The changes are shown in the abstract in the latexdiff document.**

*33. The joint effect of current- and depth-induced refraction at intermediate depth was also demonstrated by Romero et al. 2020 (Figure 14d).*

**That is a good point. We have added this reference alongside with the Halsne et al. 2022 citation. The changes are shown in Section 1 in the latexdiff document.**

*57. \chi' was already defined as \chi*

**We realize that we were not clear in the initial manuscript. \chi' is defined for the absolute angular frequency \omega, while \chi is for the intrinsic frequency \sigma. Therefore, we still stick with both the definitions. We however**

**reformulated the sentence introducing \chi' in order to clarify that we now consider \chi', which can be seen in Section 2.1 in the latexdiff document.**

*Equation (9) assumes that the currents are stationary (see for example equation 2 of Mathiesen 1987). You should either state that explicitly or account for the current acceleration.*

**Thanks for pointing this out. We have added the equation for the non-stationary current version (however, still with a fixed bathymetry), and explicitly stated that it reduces to equation 10 (in the revised manuscript) for stationary conditions. The changes are shown in Section 2.1 in the latexdiff document.**

*Equations 22-24: I suggest changing fadv to $f_{\bf{x}}$. Similarly, I also suggest changing fconc to $f_{\bf{k}}$ in equations 25-27.*

**We find this to be a good suggestion by the reviewer, and have changed the naming of the two functions accordingly, as shown in Section 3.1 in the latexdiff document.**

*119-122: I would remove the comments about the concertina effect referring to the kinematical conservation equation.*

**We agree with the reviewer and have removed these comments, as shown in Section 3.1 in the latexdiff document.**

*265: Kenyon (1971) showed this first.*

**True. We have added the Kenyon reference in addition to the Dysthe (2001) reference, as shown in Section 4.1 in the latexdiff document.**

*341-346: Again, Romero et al. 2020 carried out experiments of ray tracing with combined depth and current gradients and without currents.*

**Thanks for pointing this out. We reformulated the sentence and added a reference to the work by Romero et al. 2020 and the relevant figure therein. The changes are shown in Section 6 in the latexdiff document.**

*References*

*Kenyon, K. E., 1971: Wave refraction in ocean currents. Deep Sea Res., 18, 1023–1034.*

*Mathiesen, M., 1987: Wave refraction by a current whirl. J. Geophys. Res., 92, 3905–3912.*

*Romero, L., D. Hypolite, and J. C. McWilliams, 2020: Submesoscale current effects on surface waves. Ocean Model., 153, 101662*

**Anonymous Reviewer #2**

This is a short but useful contribution. The paper is well written. I have only one minor comment: the recent paper "Theoretical and applied considerations in depth-integrated currents for third-generation wave models" in AIP Advances (https://doi.org/10.1063/5.0077871) is related to the present note.

**Thanks for pointing us to this recent paper. We have added a paragraph in Section 6 "Discussion and concluding remarks" (see the latexdiff document), which discusses the impact of vertically sheared currents on the absolute frequency. Here we present a possible solution to the problem within the ocean_wave_tracing framework by computing the effective, depth integrated, current prior to the ray tracing computation. Doing so, however, and as highlighted in the paper mentioned by the reviewer, we also mention that caution must be exerted in the vertical integration as numerical errors can be introduced if the vertical resolution of the 3D current fields is coarse.**